# Estradiol Enhances Anorectic Effect of Apolipoprotein A-IV through ERα-PI3K Pathway in the Nucleus Tractus Solitarius

**DOI:** 10.3390/genes11121494

**Published:** 2020-12-12

**Authors:** Min Liu, Ling Shen, Meifeng Xu, David Q.-H. Wang, Patrick Tso

**Affiliations:** 1Department of Pathology and Laboratory Medicine, University of Cincinnati College of Medicine, Cincinnati, OH 45237, USA; shenln@ucmail.uc.edu (L.S.); xume@ucmail.uc.edu (M.X.); tsopp@ucmail.uc.edu (P.T.); 2Department of Medicine and Genetics, Division of Gastroenterology and Liver Diseases, Marion Bessin Liver Research Center, Einstein-Mount Sinai Diabetes Research Center, Albert Einstein College of Medicine, Bronx, NY 10461, USA; david.wang@einsteinmed.org

**Keywords:** estradiol, apolipoprotein A-IV, PI3K/Akt signaling, food intake, nucleus tractus solitarius

## Abstract

Estradiol (E2) enhances the anorectic action of apolipoprotein A-IV (apoA-IV), however, the intracellular mechanisms are largely unclear. Here we reported that the phosphatidylinositol 3-kinase (PI3K)/Akt signaling pathway was significantly activated by E2 and apoA-IV, respectively, in primary neuronal cells isolated from rat embryonic brainstem. Importantly, the combination of E2 and apoA-IV at their subthreshold doses synergistically activated the PI3K/Akt signaling pathway. These effects, however, were significantly diminished by the pretreatment with LY294002, a selective PI3K inhibitor. E2-induced activation of the PI3K/Akt pathway was through membrane-associated ERα, because the phosphorylation of Akt was significantly increased by PPT, an ERα agonist, and by E2-BSA (E2 conjugated to bovine serum albumin) which activates estrogen receptor on the membrane. Centrally administered apoA-IV at a low dose (0.5 µg) significantly suppressed food intake and increased the phosphorylation of Akt in the nucleus tractus solitarius (NTS) of ovariectomized (OVX) rats treated with E2, but not in OVX rats treated with vehicle. These effects were blunted by pretreatment with LY294002. These results indicate that E2’s regulatory role in apoA-IV’s anorectic action is through the ERα-PI3K pathway in the NTS. Manipulation of the PI3K/Akt signaling activation in the NTS may provide a novel therapeutic approach for the prevention and the treatment of obesity-related disorders in females.

## 1. Introduction

Considerable evidence suggests that estrogen, especially 17β-estradiol (E2), inhibits energy intake through increasing the potency of satiation factors, such as apolipoprotein A-IV (apoA-IV) [1]. In the present study, we focused on the nucleus tractus solitarius (NTS) as a target area because our previous studies have demonstrated that the NTS is critical to E2’s anorectic action [2] and that apoA-IV acts in this area to reduce food intake in ovariectomized (OVX) rats. Specifically, intra-4th-ventricular (i4vt) administration of low dose of apoA-IV reduced food intake more in E2-treated ovariectomized (OVX) rats than that in vehicle-treated controls [3]. However, the underlying mechanisms of how E2 enhances apoA-IV’s anorectic action are still unclear. 

Phosphatidylinositol-4,5-bisphosphate 3-kinases (PI3Ks) are a family of enzymes involved in the regulation of cellular functions, many of which are related to the capability of PI3K to activate Akt, a serine/threonine kinase that mediates PI3K downstream effects. In recent years, it has been found that PI3K/Akt signaling in the brain is involved in the control of energy homeostasis. It has been reported that E2 regulates the expression of PI3K subunits in the hypothalamus, and the PI3K/Akt cascade mediates E2’s actions in hypothalamic neurons [4]. Therefore, central PI3K/Akt signaling may become a common path that provides a coordinated control of energy balance. 

The goal of the present studies was to test the hypothesis that E2 enhances apoA-IV anorectic action through stimulating the activation of PI3K/Akt signaling pathway in the NTS of female rodents. We first explored if the PI3K/Akt pathway is regulated by E2 and/or apoA-IV in cultured primary neuronal cells isolated from rat embryonic brainstems. In the subsequent studies, we tested whether the PI3K/Akt pathway is synergistically activated in the neuronal cells which were treated with a combination of E2 and apoA-IV at their subthreshold doses. We further determined which estrogen receptor mediates E2-induced activation of the PI3K/Akt pathway. Finally, we assessed whether the 4th ventricle (i4vt) administration of LY294002, a PI3K selective antagonist, diminished E2’s actions on apo A-IV hypophagic effect and the activation of PI3K/Akt signaling pathway in the NTS. 

## 2. Materials and Methods 

### 2.1. Animals

Long-Evans pregnant-timed (day-18) rats (Harlan, Indianapolis, IN, USA) were used for obtaining neuronal cells from brainstem. Female Long-Evans rats (10-week old, from Harlan) were individually housed in a temperature-controlled vivarium. The rats were provided with standard rodent diet and water ad libitum except where noted. All procedures used in the experiments were approved by the Institutional Animal Care and Use Committee of the University of Cincinnati.

### 2.2. Materials

17β-estradiol (E2), E2-BSA (E2 conjugated to bovine serum albumin) were purchased from Sigma (St. Louis, MO, USA). LY294002 (a PI3K inhibitor), monoclonal antibodies anti-pAkt and anti-Akt from rabbits were obtained from Cell Signaling Technologies Inc. (Beverly, MA). PPT (4,4′,4″-(4-propyl-[1H]-pyrazole-1,3,5-triyl) trisphenol, an ERα agonist) and DPN (2,3-*bis*(4-hydroxyphenyl)-propionitrile, an ERβ agonist) were bought from Tocris (Ellisville, MO, USA). Cell culture medium and other chemicals were obtained from Thermo Fisher Scientific (Waltham, MA, USA) and Sigma, respectively. 

### 2.3. Culture and Treatment of Primary Cultured Neuronal Cells

As we reported previously [5], primary neuronal cells from rat brainstem were cultured in a neurobasal medium, which was serum-free and supplemented with B-27. Briefly, brainstems were dissected from the fetal brains and placed in Hanks’ balanced salt solution. The tissues were dispersed by repeated pipetting. After being settled for 3 min, the supernatant was transferred to a fresh tube and centrifuged at 200× *g* for 2 min. The pellet was resuspended in a serum-free neurobasal medium supplemented with B-27, plus with 25 μM l-glutamic acid and 0.5 mM l-glutamine (Invitrogen). In 24-well plates, the cells were plated at a density of 5 × 10^4^ cells/well and cultured at 37 °C. After 3-day incubation, we replaced half of the medium with fresh medium with cytosine arabinofuranoside (10 μM; Sigma), but without l-glutamic acid at that time. Under these conditions, less than 5% glial cells existed in the cultured cells, which was confirmed by glial marker-staining [6].

Six days after the culture, the neuronal cells were incubated with LY-294002, E2, PPT, and DPN for different time points, followed by protein isolation and immunoblotting, as we did before [6]. These chemicals were dissolved in analytically pure dimethylsulfoxide (DMSO) first, and then diluted in culture medium. The final concentrations of 10 μM LY-294002, and 10 nM E2, PPT , DPN were based on previous reports [4,5,7]. 

### 2.4. Surgical Procedure for i4vt Cannula Implantation

The procedure was described as we reported previously [3]. Briefly, after being anesthetized, rat was placed in a stereotaxic instrument and a stainless-steel guide cannula (24-gauge from Plastics One, Roanoke, VA, USA) was implanted. The stereotaxic coordinates for the implanted guide cannula were (relative to occipital suture): 2.5 mm posterior, on the midline of the skull, and 4.5 mm ventral from the dura [3,8]. Cannulas were attached to the skull with dental acrylic and sealed with an obturator. Seven days later, the animals were removed from the stereotaxic instrument and housed individually. Cannula placement was verified by measuring the hyperglycemic response to 5-thio-d-glucose (210 μg in 2 μL of saline), which is a nonmetabolizable glucose isomer causing counter-regulatory responses in NTS neurons and increased glucose levels in plasma [9]. Rats responding with an increase of glucose concentration in plasma by more than 100% after 30 min were used [3,8].

### 2.5. Ovariectomy (OVX) 

Overnight-fasted rats with i4vt cannulas were anesthetized and then underwent bilateral OVX through a midline abdominal incision, as we described previously [3]. 

### 2.6. pAkt/Akt Levels Determined by Immunoblot Analyses

As we reported previously [10], the cultured neuronal cells or NTS tissues were homogenized and proteins were extracted. The proteins (10 μg) per sample were separated by SDS-PAGE and transferred onto nitrocellulose membranes (Bio-Rad, Hercules, CA). Both pAkt and Akt antibodies were diluted in 1:1000, and the same membranes incubated with phospho-antibody were stripped and reprobed with non-phospho-antibody for the comparison of phosphorylated and total AKT protein. SuperSignal West Pico Chemiluminescence reagents (Thermo Fisher Scientific, Waltham, MA, USA) were used to visualize the bands on X-ray films and NIH Image J software (http://rsb.info.nih.gov/ij/) was used to quantify the intensity of the bands. In each individual sample, the phosphorylation level of the Akt was normalized to the level of total Akt protein and presented as a ratio [10]. 

### 2.7. Statistics

All results are reported as mean ± standard error (S.E.). Statistical analysis was performed using GraphPad Prism 8 software. In vitro experiments were conducted on at least three separate occasions and each test was run in duplicate. Differences among the groups in the in vitro and the in vivo studies were determined using one-way or two-way ANOVA analyses, which were followed by Tukey test. A *p* value of 0.05 was used as the threshold for statistical significance. 

## 3. Results

### 3.1. Akt Was Phosphorylated by E2 in Cultured Neuronal Cells 

To determine the time-course changes in the activation of PI3K signaling pathway induced by E2, the cultured primary neuronal cells were treated with E2 at a concentration of 10 nM for 0, 5, 15, 30, and 60 min, respectively. As depicted in Figure 1A,B, E2 increased Akt phosphorylation in a rapid and transient manner (treatment among time points: F_[4, 8]_ = 15.70, *p* < 0.01). A significant increase in the phosphorylation of Akt level was evident at 5 min after E2 treatment, and then reached the maximum at 30 min, compared to that at 0 min. E2-induced Akt phosphorylation dropped down by 60 min (Figure 1A,B). No significant difference was found in total Akt levels under different treatments. Therefore, the total Akt was used as an internal control for protein loading of each sample in the immunoblotting analysis in all following experiments.

### 3.2. E2 Concentration-Dependently Increased Akt Phosphorylation

The cultured neuronal cells were treated with three different concentrations of E2 (1, 10, and 100 nM) for 30 min. When E2 concentration were ≥10 nM, it significantly increased Akt phosphorylation (Figure 1C,D), compared to vehicle (DMSO) treatment (treatment among concentrations: F_[3, 9]_ = 19.84, *p* < 0.01). While 100 nM E2 is 10 times higher than 10 nM E2, no marked increase in pAkt level was found, therefore, 10 nM E2 was used to treat cultured cells for 30 min in the following experiments.

### 3.3. E2-Induced Phosphorylation of Akt Was Attenuated after Pretreatment of PI3K Inhibitor

To determine if Akt was phosphorylated by E2-activated PI3K signaling pathway, the cultured cells were pretreated with LY294002 (10 μM) or vehicle (DMSO) for 30 min. The cells were then treated with E2 (10 nM) or vehicle (DMSO) for an additional 30 min. We found that, while treatment with PI3K inhibitor alone did not significantly affect pAkt level in the cultured neuronal cells, the LY294002 was able to significantly attenuate E2-induced increase in Akt phosphorylation (treatment among groups: F_[3, 6]_ = 18.96, *p* < 0.01) (Figure 1E,F), compared to vehicle-treated cells.

### 3.4. E2-Induced Activation of PI3K/Akt Pathway Was Mediated through ERα

To identify which estrogen receptor (ER) subtypes, ERα or ERβ, mediates E2’s regulation of PI3K activity, E2, ERα agonist (PPT), or ERβ agonist (DPN) were added, respectively, into the medium at the final concentration of 10 nM, and the neuronal cells were then cultured for an additional 30 min [11]. As shown in Figure 2A,B, compared to vehicle control, the PPT induced a significant increase in pAkt level, which was comparable to that after E2-treatment (treatment among groups: F_[3, 6]_ = 16.36, *p* < 0.01), whereas the DPN did not show a marked change.

### 3.5. Membrane-Associated ER Mediated E2-Induced PI3K Activation

To further investigate whether E2 regulates PI3K activation through a nongenomic mechanism, the cultured cells were treated with E2 or E2-BSA (10 nM) for 30 min. As shown in Figure 2C,D, the pAkt level induced by E2-BSA was comparable to that induced by E2 (treatment among groups: F_[2, 6]_ = 13.79, *p* < 0.01), implying that the primary mediator of E2’s action on PI3K activation was membrane-associated ERα.

### 3.6. Effects of apoA-IV Treatment on the Phosphorylation of Akt

To determine whether apoA-IV also affects the activation of PI3K/Akt signaling pathway, pAkt levels were measured in cultured neuronal cells following apoA-IV treatment. As depicted in Figure 3A,B, the phosphorylation of Akt was significantly increased after apoA-IV (100 nM) treatment at both 15 and 30 min, compared to that at the time before the treatment with apoA-IV (treatment among time points: F_[4, 8]_ = 14.22, *p* < 0.01). The increased Akt phosphorylation induced by apoA-IV completely disappeared at 60 min.

The neuronal cells were treated with apoA-IV for 15 min at different concentrations (10, 100, and 1000 nM) [12]. Compared to vehicle treatment, apoA-IV 100 nM or higher significantly increased the levels of pAkt in those cells (treatment among concentrations: F_[3, 9]_ = 16.82, *p* < 0.01) (Figure 3C,D), although no difference was found at a low concentration of apoA-IV (10 nM).

To test if apoA-IV-induced phosphorylation of Akt is blocked or attenuated by pharmacological inhibition of the PI3K signaling pathway, the primary cells were pretreated with LY294002 (10 μM) or vehicle (DMSO) for 30 min, and then treated with apoA-IV (100 nM) or vehicle (PBS) for an additional 15 min. As shown in Figure 3E,F, the effect of apoA-IV on the phosphorylation of Akt (treatment among groups: F_[3, 6]_ = 13.81, *p* < 0.01) was blocked by LY294002 (Figure 3E,F).

### 3.7. Synergistic Interaction of E2 and apoA-IV in PI3K/Akt Activation of Neuronal Cells

To test the hypothesis that E2 interacts with apoA-IV to activate the PI3K/Akt signaling pathway in neuronal cells, five groups were included, with each receiving a different combination of three components. Those combinations included (1) DMSO (vehicle for LY294002) + DMSO (vehicle for E2) + PBS (vehicle for apoA-IV); (2) DMSO + E2 (1 nM, a subthreshold dose) + PBS; (3) DMSO + DMSO + apoA-IV (10 nM, a subthreshold dose); (4) DMSO + E2 (1 nM) + apoA-IV (10 nM); (5) LY294002 (10 µM) + E2 (1 nM) + apoA-IV (10 nM). The time for LY294002 or its vehicle (DMSO) treatment was 30 min, followed by E2 or its vehicle (DMSO) treatment for 15 min, and then apoA-IV or PBS for an additional 15 min. As depicted in Figure 4, although individual subthreshold dose of E2 or apoA-IV did not markedly affect the activation of PI3K/Akt signaling pathway, the combined subthreshold doses of E2 (1 nM) and apoA-IV (10 nM) significantly increased pAkt levels (treatment among groups: F_[4, 8]_ = 8.52, *p* < 0.01). More importantly, this synergistic effect was completely blocked by the pretreatment with LY294002.

### 3.8. Central Inhibition of PI3K/Akt Signaling Significantly Diminished apoA-IV’s Anorectic Action in E2-Treated OVX Rats

To examine whether PI3K/Akt signaling activation in the brainstem mediates E2-enhanced apoA-IV’s effect on food intake, the OVX rats with i4vtcannula received a subcutaneous injection of 4 µg E2 or 0.1 mL vehicle (sesame oil) at 1000 h [13]. The dose of E2 (4.0 µg) based on previous report [13]. On the 2nd day, 1 nM LY294002 or vehicle (1 µL, 10% DMSO in artificial CSF, aCSF) was i4vt injected 1 h prior to dark phase. The dose of LY294002 was chosen was based on previous report that it did not affect food intake when administered alone, but diminished leptin-induced inhibition of food intake [14] and insulin-induced reduction of blood glucose [15]. Sixty minutes later, the OVX rats received i4vt injection of 0.5 μg apoA-IV or aCSF (1 µL) at dark onset. The selected apoA-IV dose was based on our previous report that it significantly reduced food intake only in E2-treated, but not vehicle-treated, OVX rats [3].

In either oil or E2-treated OVX rats, there were four counterbalanced conditions: control group (10% DMSO followed by aCSF), LY294002 group (LY294002 followed by aCSF), apoA-IV group (10% DMSO followed by apoA-IV), and the combination group (LY294002 followed by apoA-IV). At the onset of the dark, the OVX rats were provided with pre-weighed chow, and the food intake was measured at 45 min after food presentation. Food intake and body weight was also recorded at 24 h after refeeding. 

As depicted in Figure 5A, the food intake at 45 min was significantly reduced after treatment with apoA-IV at a low dose of 0.5 µg only in E2-treated, but not in vehicle (oil)-treated, OVX rats, compared with that after aCSF (vehicle) treatment (treatment among groups: F_[3, 48]_ = 4.309, *p* < 0.05). No significant difference was observed in food intake and body weight at 24 h in rats receiving either apoA-IV or aCSF injections. Importantly, pretreatment with LY294002 significantly attenuated the potency of exogenous apoA-IV to induce satiation, although LY294002 treatment alone did not considerably affect food intake in OVX rats.

To further determine if the activation of PI3K/Akt pathway is necessary for the reduction of food intake induced by E2 and apoA-IV, after recovery from the feeding test, the same animals were fasted for 4 h, and then received the same dose of i4vt administration of LY294002 or vehicle, followed by apoA-IV or aCSF injection in E2 or oil-treated OVX rats, as described above. At 30 min after the 2nd injection, we sacrificed animals and rapidly dissected brains, which were frozen at −80 °C. NTS were micropunched from coronal brain sections cut using a cryostat. As we reported previously [3], the total proteins extracted from the NTS were used for immunoblot analysis.

As depicted in Figure 5B,C, apoA-IV significantly stimulated the phosphorylation of Akt in OVX rats receiving E2-treatment, but not oil-treatment, compared to vehicle (aCSF). More interestingly, this effect was significantly attenuated by pretreatment with LY294002 (treatment among groups: F_[3, 48]_ = 5.036, *p* < 0.05). No significant change in pAkt level was observed when LY294002 was administered alone.

## 4. Discussion

E2 potently reduces energy intake [16,17]. In the rodents, females consume different amounts of food across their ovarian cycles (normally 4–5 days). Specifically, the females eat the most during diestrus when E2 level is relatively low, and the least during estrus, which occurs right after the preovulatory rise in E2 secretion [18]. Disruption of estrous cycling through OVX results in the increase of food intake and body weight gain. Interestingly, a cyclic replacement of E2 at a physiologic dose eliminates hyperphagia induced by OVX and reduces body weight to the levels of intact rodents gonadally [3,18]. Although E2’s anorectic action is well-known, the underlying mechanisms are still largely unclear.

E2 acts in the brain to reduce food intake [19], and compelling evidence suggests that the NTS is an important brain area for E2’s anorexigenic action [2,17,20]. In the NTS, ERα is highly expressed [21], and the administration of E2 at a very low dose onto the surface of the hindbrain over the NTS significantly reduced food consumption and activated neurons that contain ERα in the NTS [2]. These results support our hypothesis that the NTS is the brain site, where E2 interacts with apoA-IV to control of food intake.

It has been reported that E2 controls food intake through increasing the potency of satiation factors [1].Our previous studies have demonstrated that ApoA-IV is one of satiation factors. In rodents, central administration of apoA-IV dose-dependently reduces food intake. Meal size was increased when an specific antibody was used to block endogenous apoA-IV’s action, suggesting that endogenous apoA-IV tonically inhibits food intake [22,23]. More importantly, the anorexigenic action of apoA-IV in OVX rats is enhanced by E2 treatment [3], implying that E2 may enhance apoA-IV’s effect on food intake through unknown signaling pathway.

E2’s effects can be divided into genomic and nongenomic, and these are mediated by specific estrogen receptors (ERs) [24]. In its genomic action, E2 binds to ERα or ERβ and then induces ER homodimerization and subsequent nuclear translocation. Within the nucleus, the ER binds to estrogen response elements (EREs) located in the promoter regions of target genes, resulting in the recruitment of specific coregulators (coactivators or corepressors) and increased or decreased target gene transcription [25]. Our previous studies have demonstrated that E2 stimulates *apoA-IV* gene expression through ERα in the NTS of OVX rodents [3,5].

Nongenomic effects of E2 are mediated by membrane subpopulations of ERα and ERβ, and these effects are frequently associated with the activation of various protein-kinase cascades [26]. Considering apoA-IV’s anorectic effect is relatively acute in E2-treated OVX rats, it seems unlikely that the reduction of food intake is regulated through transcriptional events, which typically require longer time. A signaling cascade, such as PI3K/Akt, is more likely responsible for the decreased food intake. Previous studies suggest that PI3K signaling is important in E2-mediated actions in the brain. For example, E2 increases the central level of insulin-like growth factor 1 (IGF-1) through the activation of PI3-K/Akt [27]. Additionally, E2-induced neuronal survival against toxic effects from Aβ_1–42_ is also related to the activation of PI3K/Akt pathway [28].

In the present study, we demonstrated that E2 stimulated the activation of PI3K/Akt signaling pathway in a time- and concentration-dependent manner in primary brainstem neuronal cells, and this effect was significantly attenuated by LY294002, a pharmacological inhibitor of PI3K signaling pathway (Figure 1). Although there are several estrogen receptors (ERs), ERα has been reported as a main mediator for E2’s actions on energy homeostasis [13]. Therefore, it is more likely that E2 acts through ERα to increase the activation of this signaling pathway in the neurons. This possibility was tested by the treatment of E2, PPT (ERα agonist), or DPN (ERβ agonist), respectively, in cultured neuronal cells. We found that E2 and PPT, but not the DPN, exerted comparable effects, indicating that ERα played an important role in the activation of PI3K signaling pathway in the neuronal cells (Figure 2).

To further determine whether ERα on cell transmembrane is involved in E2mediated PI3K/Akt signaling activation, the cultured neuronal cells were treated with E2-BSA. Due to its large size and charge properties, the E2-BSA only interacted with ERs on the cell membrane [29]. Interestingly, the pAkt level in E2-BSA-treated cells was comparable to that in E2-treated cells (Figure 2). These results indicate that the ERα on the membrane is the primary mediator of E2’s effect on PI3K/Akt signaling activation.

We further investigated whether PI3K/Akt signaling pathway is also activated by apoA-IV through examining the phosphorylation of Akt in neuronal cells after apoA-IV treatment. We found that apoA-IV increased pAkt levels time- and concentration-dependently, and this effect was attenuated by LY294002 treatment (Figure 3).

Although E2 or apoA-IV individually activated PI3K/Akt signaling pathway, it is unclear if they work together. To test this possibility, the cultured neuronal cells were treated with subthreshold doses of E2 and apoA-IV. It is found that such combination significantly increased pAkt levels. More importantly, this synergistic effect was completely blocked by the pretreatment with LY294002 (Figure 4). These results clearly demonstrated that E2 interacted with apoA-IV to activate PI3K/Akt signaling in the neuronal cells.

The mechanisms by which E2 enhances apoA-IV anorectic action in the NTS is still unclear. Since both E2 and apoA-IV activate the PI3K pathway in cultured neurons isolated from brainstem, the PI3K cascade in the NTS may become a common pathway that integrates the signals of E2 and apoA-IV, leading to enhanced anorectic effect. To test this possibility, we administered LY294002, followed by apoA-IV into the 4th-ventricle of OVX rats treated with E2 or vehicle, respectively. The treatment with E2 in OVX rats significantly enhanced the food-suppressive effect of apoA-IV and increased Akt phosphorylation in the NTS. Importantly, the inhibitory effect on food intake induced by apoA-IV was significantly diminished by LY294002. ApoA-IV-induced phosphorylation of Akt was also significantly attenuated by LY294002 (Figure 5). These data collectively suggested that PI3K activation in the NTS is necessary for E2-enhanced effect on apoA-IV-induced anorexia.

It should be noted that, except for the ERα on the membrane, E2 also acts on a membrane-bound cytosolic ER, such as G protein-coupled estrogen receptor 1 (GPER1, also known as GPR30), which is localized mainly in the endoplasmic reticulum [30,31]. Recent studies found that central treatment with G1, the GPER1 agonist, reduces food intake and body weight gain in WT mice [32], and GPER1 KO mice are obese with decreased energy expenditure [33,34]. Whether E2 interacts with apoA-IV to reduced food intake through GPER1 should be addressed in the future.

Here, we present out study on the neuronal cells, and it is worth pointing out that the effects induced by the interaction of E2 and apoA-IV could also occur in astrocytes. Astrocytes are the most abundant glial cells in the mammalian brain. At very beginning, these cells are described as passive supporters of neurons [35], however, data in subsequent years have revealed that glial cells are actively involved and required for effective function of the CNS [36]. Interestingly, astrocytes are targets for estrogen action [37] as reflected by the fact that astrocytes express both ERα and ERβ receptors either on their plasma membranes or intracellularly [38]. Activation of the membrane estrogen receptor initiates a rapid, free cytoplasmic calcium concentration ([Ca^2+^]) flux via the phospholipase C (PLC)/inositol trisphosphate (IP3) pathway [39]. In our previous study, we also revealed that apoA-IV was presented in the astrocytes [40]. Whether the combination of E2 and apoA-IV affects the PI3K/Akt signaling pathway in astrocytes should also be tested in the future study.

While we focused on dietary obesity in female rats in the present study, the fundamental knowledge obtained from our findings will allow us to address other clinical problems in women; i.e., anorexia nervosa. Why are women more at risk for eating disorders? Is the development of these disorders related to altered PI3K/Akt signaling pathway in the brain? If so, is it possible to attenuate the development of eating disorders by regulating central PI3K/Akt signaling pathway? Finding the answers for these questions are significant because severe eating disorders produce serious psychological and physical health problems, and millions of women are suffering from them in the USA [41].

## 5. Conclusions

The present studies for the first time demonstrated that E2 interacted with apoA-IV to significantly increase PI3K/Akt signaling activation in cultured primary neuronal cells. It was further confirmed by that 4th ventricle-delivery of a PI3K inhibitor significantly attenuated apoA-IV-induced effects on food intake and the activation of PI3K/Akt signaling pathway in the NTS of E2-treated OVX rats. These results suggested a novel mechanism, through which E2 positively regulated apoA-IV’s anorectic action in the NTS. Manipulation of the activation of PI3K/Akt cascade in the NTS may provide a novel therapeutic approach for the prevention and the treatment of obesity-related disorders in females.

## Figures and Tables

**Figure 1 genes-11-01494-f001:**
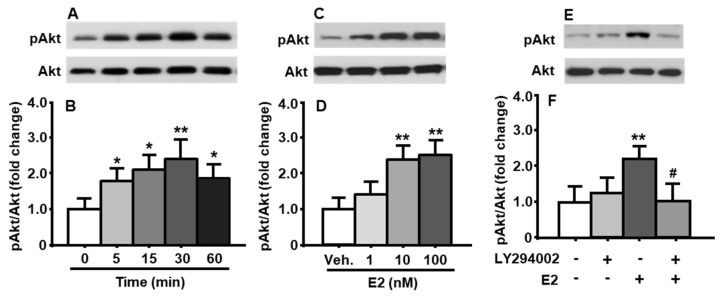
PI3K/Akt signaling pathway in cultured neuronal cells was significantly activated by E2. E2 at a concentration of 10 nM time-dependently increased the phosphorylation of Akt (**A**,**B**). When treated for 30 min, E2 concentration-dependently increased the ratio of pAkt/total Akt (**C**,**D**). PI3K inhibitor, LY294002 (10 μM), significantly attenuated E2-induced Akt phosphorylation (**E**,**F**). Representative immunoblots of pAkt and total Akt are shown in (**A**,**C**,**E**); quantitative analysis of the data is presented in (**B**,**D**,**F**). Veh.: vehicle (DMSO). Mean ± S.E., n = 3–4. ** p* < 0.05, *** p* < 0.01 compared to pAkt levels at 0 min or vehicle control. *^#^ p* < 0.05, compared to that in neuronal cells treated with E2 for 15 min.

**Figure 2 genes-11-01494-f002:**
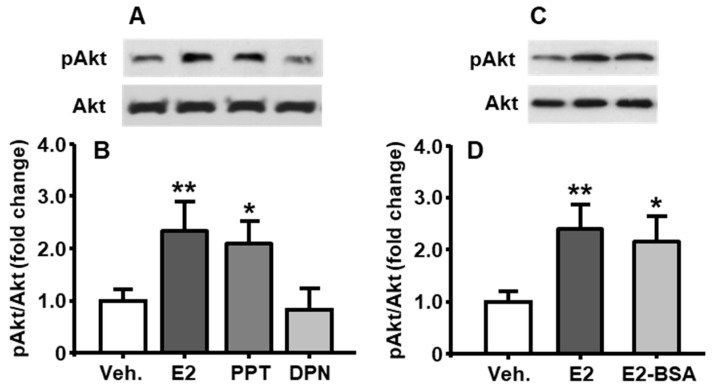
Comparison of PI3K/Akt pathway activation after different treatments in cultured neuronal cells. E2, PPT, but not DPN (all at 10 nM), significantly increase pAkt levels at 30 min (**A**,**B**). E2-BSA (10 nM) also increased the phosphorylation of Akt (**C**,**D**). Mean ± S.E., n = 3–4. * *p* < 0.05, ** *p* < 0.01 compared to pAkt level after vehicle treatment.

**Figure 3 genes-11-01494-f003:**
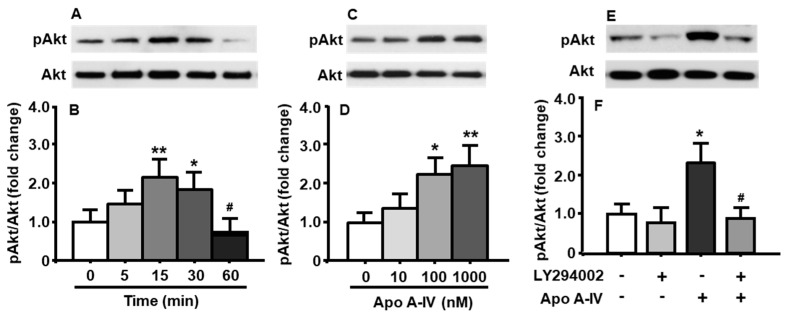
ApoA-IV activated PI3K/Akt signaling pathway in cultured neuronal cells. ApoA-IV (100 nM) time-dependently increased the phosphorylation of Akt (**A**,**B**). When treated for 15 min, apoA-IV concentration-dependently increased the ratio of pAkt/total Akt (**C**,**D**). This effect was significantly attenuated by LY294002 (10 μM) (**E**,**F**). Mean ± S.E., n = 3–4. * *p* < 0.05, ** *p* < 0.01, compared to pAkt level at 0 min or vehicle control. ^#^
*p* < 0.05, compared to that in neuronal cells treated with apoA-IV for 15 min (B) and ApoA-IV treated alone (F), respectively.

**Figure 4 genes-11-01494-f004:**
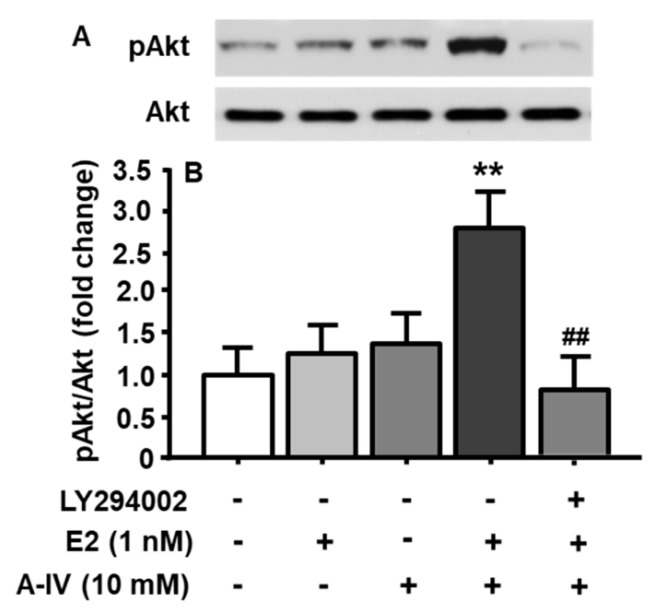
Combination of E2 and apoA-IV at their subthreshold doses significantly activated PI3K/Akt signaling pathway in cultured neuronal cells. Representative immunoblots of pAkt and total Akt are shown in (**A**), and quantitative analysis of the data is presented in (**B**). Mean ± S.E., n = 3. ** *p* < 0.01, compared to pAkt level in the cells treated with vehicles. ^##^
*p* < 0.01, compared to pAkt level in the cells treated with the combination of E2 and apo A-IV.

**Figure 5 genes-11-01494-f005:**
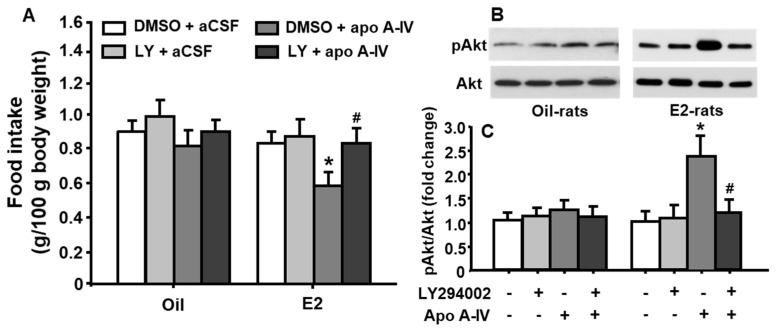
Central apoA-IV (0.5 µg) administration resulted in significant inhibition of food intake, and significant increase of Akt phosphorylation in the NTS of E2-treated OVX rats. These effects were attenuated by LY294002. (**A**) Changes in food intake after apo A-IV or vehicle (aCSF) for 45 min in oil- or E2-treated OVX rats. (**B**) Representative immunoblots of pAkt and total Akt; (**C**) quantitative analysis of the data. Means ± SE, n = 7 per group, * *p* <0.05, vs. oil-OVX rats treated with vehicle, and ^#^
*p* <0.05, vs. E2-OVX rats treated with apoA-IV.

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
