# Peer review of "Estradiol Enhances Anorectic Effect of Apolipoprotein A-IV through ERα-PI3K Pathway in the Nucleus Tractus Solitarius"

_genes, 2020, doi:10.3390/genes11121494_

Round 1

Reviewer 1 Report

This papers explores the role of oestrogen signalling on the effect of Apoapolipoprotein A-IV in energy intake.  The paper further explores how E2 functions through ER-alpha and how this acts synergistically with apolipoprotein A-IV and that this signalling is dependent upon PI3K/AKT signalling.  The experiments are sound and relevant to the biological question. 

This work provides further data on a novel therapeutic approach for the treatment of obesity in females.  Although it should be noted that the aromatisation of testosterone to oestrogen in males does not rule out the therapeutic potential in males - providing a potential further avenue of research. 

Minor comments on the manuscript: 

Results: Figure 1 a) The results section referring to this figure discusses that overall AKT levels are unchanged - how has this been measured? If through analysis of AKT compared to loading control (beta actin etc..) this should be mentioned and shown in the text and figure. 

Figure 5: It is unclear from the text or methods if in behavioural experiments investigating food intake were animals house individually and each n. number therefore a separate cage.  Please update text to reflect experimental design. 

General comments:

Please update figure legends with concentrations of each pharmacological agent used - this is clear in the dose dependent experiments and suboptimal dose experiments (figure 4) but not single concentration experiments. 

Discussion: Should include discussion of the potential role if any in males - please see comment above at start of comments. 

Reviewer 2 Report

This study attempts to identify a novel mechanism through which estradiol exerts its anorexigenic effects in female rats. The authors hypothesized that estradiol enhances the effect of the satiety signal, apolipoprotein A-IV, and that this occurs through activation of the PI3K signaling pathway in the NTS. They utilized both in vitro and in vivo models to investigate this question and the experiments were well-designed and clearly communicated and the results appear relatively straight-forward. I have a few concerns/points of clarification and some suggestions to enhance the discussion of these interesting data:

  1. Please report all F values and degrees of freedom in the results section and state whether there were, in fact, significant interactions in your experiments that utilized a 2x2 design and thus analyzed with a two-way ANOVA.
  2. Please report sample sizes for each group for in vivo experiments.
  3. On page 4, line 154-155, I believe the authors mean to state, “to determine if phosphorylation of Akt was mediated by the PI3K signaling pathway, the primary cells were pretreated with LY294002.” As it’s written, it reads the opposite, that PI3K is signaling is mediated/controlled by Akt phosphorylation, which is downstream of PI3K. This was correctly written on page 5, line 193-194.
  4. Have the primary neuronal cultures been phenotyped to delineate the types of neurons being cultured/treated? For example, the NTS houses many neuronal phenotypes important for the control of food intake and regulation of energy balance, namely GLP-1 neurons as well as a subset of POMC neurons, both of which express multiple estrogen receptor subtypes.
  5. Related to my previous comment, which types of neurons respond to apoA-IV? Does apoA-IV act in the hypothalamus to reduce food intake? If so, this could be a good follow-up study given that estradiol also acts in the hypothalamus (ARC, PVN, LH) to suppress food intake. These points could be added to the discussion.
  6. The authors make no mention of the G protein-coupled estrogen receptor, GPER-1. While it is true that ERalpha appears to be the main receptor responsible for estradiol’s effect on food intake, multiple studies have also shown GPER-1 is sufficient to reduce food intake in rats, mice, and guinea pigs (see: https://doi.org/10.1016/j.yhbeh.2018.05.018 or https://doi.org/10.1159/000338669). GPER-1 is also expressed in the NTS (see: https://doi.org/10.1016/j.physbeh.2015.05.032) and may be critical for PPT’s effect on food intake (see: https://doi.org/10.1016/j.yhbeh.2018.05.018). Interestingly, GPER-1 may also be associated with PI3K signaling, at least in the hippocampus (see : https://doi.org/10.1016/j.neuroscience.2016.04.026). These points should be added to the discussion and could also be good follow-up studies to get an even better idea of what is happening in the NTS with estradiol.

Reviewer 3 Report

In the current manuscript, the author showed that estradiol enhances the anorectic effect of apolipoprotein A-IV through ERα-PI3K pathway in the nucleus tractus solitarius. In my opinion, the manuscript does not fit in the scope of the GENES journal.

Major concern:

 The degree of novelty is reduced since there is a series of papers of the same group showing that:

  1. Apolipoprotein A-IV exerts its anorectic action through a PI3K/Akt signaling pathway in the hypothalamus, Ling Shen, Chunmin C. Lo, Laura A. Woollett, and Min Liua, Biochem Biophys Res Commun. 2017 Dec 9; 494(1-2): 152–157. PMID: 29037812
  2. Estradiol Increases the Anorectic Effect of Central Apolipoprotein A-IV, Ling Shen, David Q.-H. Wang, Chun-min Lo, Patrick Tso, W. Sean Davidson, Stephen C. Woods, and Min Liu, Endocrinology. 2010 Jul; 151(7): 3163–3168. PMID: 20484461
  3. Estradiol stimulates apolipoprotein A-IV gene expression in the nucleus of the solitary tract through estrogen receptor-α. Shen L, Liu Y, Wang DQ, Tso P, Woods SC, Liu M. Endocrinology. 2014 Oct;155(10):3882-90. PMID: 25051443

The paper looks to be a mixture of all these previous data.

It is very strange that the paper published in BBRC is not cited in the manuscript. Fig1A is so similar to Fig 1A from BBRC paper. Moreover, the upregulatory effect of E2 on apoA-IV previously revealed by the same group is not mentioned.  

The central point of the paper is the convergence of the pathways to Akt at the subliminal concentration of both E2 and apoA-IV. However, this is not a synergistic interaction of E2 and apoA-IV, as mentioned in the manuscript. This must be emphasized and the mechanism determined step by step, including how apoA-IV transmits the signals in the cells. Moreover, Akt phosphorylation depending on the balance of apoA-IV and E2 concentrations would be very useful.

Minor concerns:

For the difference significance shown in Fig.5A must be specified the reference.  

Reviewer 4 Report

This study represents a carefully performed and significant contribution to the understanding of anorexic effects of estrogen.  The results showed that estradiol enhanced the ability of ApoA-IV to stimulate the PI3K/AKT pathway in cultured cells.  Estradiol also enhanced anorexic effects of ApoA-IV infusions into the area of the dorsal medulla and provided evidence that effects of estradiol on ApoA were related to effects upon feeding. The results also suggested that effects of estradiol on membrane bound estrogen receptor type Alpha at least in part explain these results. 

The methodology employed in this study was sophisticated, up to date, and appropriate for the goals of this research.  Statistical analysis was also appropriate.

Some conclusions in the discussion should be slightly modified.  For example, the authors emphasize that their data show that membrane-bound estrogen receptors are involved in anorexic effects of estrogen.  This is true, but genomic, nuclear effects of estrogen also seem to be involved in estrogen induced anorexia.  Food intake in rats does not fall until several days after blood levels of estradiol peak during the estrus cycle, and food intake shows maximal declines only days after estradiol injections (reviewed in Rivera HM, Activation of central, but not peripheral, estrogen receptors is necessary for estradiol's anorexigenic effect in ovariectomized rats, Endocrinology . 2010 Dec;151(12):5680-8.).  These observations suggest that delayed, genomic effects of estrogen are also pertinent to this question.

The authors emphasize that synergistic effects of estradiol and ApoA-IV in nerve cells probably underlie their observations.  This indeed is the most likely explanation.  However, traces of ApoA-IV have also been detected in astrocytes (Shen L, Characterization of apolipoprotein A-IV in brain areas involved in energy homeostasis, Physiol Behav 2008 95(1-2):161-7 AND Fukagawa K, Immunoreactivity for apolipoprotein A-IV in tanycytes and astrocytes of rat brain. Neurosci Lett. 1995 Oct 13;199(1):17-20).  Also, estradiol affects many functional properties of astrocytes (see Fuente-Martin E, Estrogen, astrocytes, and the neuroendocrine control of metabolism, Rev Endocr Metab Disord  2013 14(4):331-8). 

Finally, a recent paper shows that astrocytes of the dorsal medulla influence feeding behavior (see MacDonald AJ, Regulation of food intake by astrocytes in the brainstem dorsal vagal complex, Glia. 2020 68(6):1241-1254).  Some mention of the possibility of an interaction between estradiol and ApoA-IV upon astrocytes would therefore be appropriate.

Finally, the value of this study is not limited to a better understanding of sex differences in obesity, as noted by the authors.  A disturbance in estrogen signaling pathways may also be present in feeding disorders such as anorexia nervosa, which shows a remarkable sex difference in incidence (see Cui H, Behavioral disturbances in estrogen-related receptor alpha-null mice. Cell Rep  2015 11(3):344-50 and Young JK. (2010) Anorexia nervosa and estrogen.  Current status of the hypothesis.  Neurosci. Biobehav. Rev. 34:1195-1200).

I found a small typographical error on line 222 of the manuscript (change "minute" to "minutes"
